# Colorimetric Kit for Rapid Porcine Circovirus 2 (PCV-2) Diagnosis

**DOI:** 10.3390/pathogens11050570

**Published:** 2022-05-12

**Authors:** Caroline Rodrigues Basso, Ana Carolina Yamakawa, Taís Fukuta Cruz, Valber Albuquerque Pedrosa, Massimiliano Magro, Fabio Vianello, João Pessoa Araújo Júnior

**Affiliations:** 1Biotechnology Institute, São Paulo State University, Botucatu 18607-440, SP, Brazil; a.yamakawa@unesp.com (A.C.Y.); tfcruz@yahoo.com.br (T.F.C.); joao.pessoa@unesp.br (J.P.A.J.); 2Chemical and Biological Sciences Department, Bioscience Institute, São Paulo State University, Botucatu 18618-000, SP, Brazil; valber.pedrosa@unesp.br; 3Comparative Biomedicine and Food Science Department, University of Padua, 35020 Legnaro, Italy; massimiliano.magro@unipd.it (M.M.); fabio.vianello@unipd.it (F.V.)

**Keywords:** colorimetric kit, hybrid nanoparticles and porcine circovirus 2

## Abstract

The aim of the current study is to present a low-cost and easy-to-interpret colorimetric kit used to diagnose porcine circovirus 2 (PCV-2) to the naked eye, without any specific equipment. The aforementioned kit used as base hybrid nanoparticles resulting from the merge of surface active maghemite nanoparticles and gold nanoparticles, based on the deposition of specific PCV-2 antibodies on their surface through covalent bonds. In total, 10 negative and 40 positive samples (≥10^2^ DNA copies/µL of serum) confirmed by qPCR technique were tested. PCV-1 virus, adenovirus, and parvovirus samples were tested as interferents to rule out likely false-positive results. Positive samples showed purple color when they were added to the complex, whereas negative samples showed red color; they were visible to the naked eye. The entire color-change process took place approximately 1 min after the analyzed samples were added to the complex. They were tested at different dilutions, namely pure, 1:10, 1:100, 1:1000, and 1:10,000. Localized surface plasmon resonance (LSPR) and transmission electron microscopy (TEM) images were generated to validate the experiment. This new real-time PCV-2 diagnostic methodology emerged as simple and economic alternative to traditional tests since the final price of the kit is USD 4.00.

## 1. Introduction

Porcine circovirus (PCV) is described as one of the most endemic viruses worldwide [1]. PCV viruses belong to family *Circoviridae*, genus *Circovirus*; they have icosahedral shape, 17 nm (in diameter), as well as present a single-stranded covalently closed circular DNA genome and one main capsid protein. There are four species of PCV, namely porcine circovirus 1 (PCV-1), porcine circovirus 2 (PCV-2), porcine circovirus 3 (PCV-3), and porcine circovirus 4 (PCV-4). Epidemiological studies have shown that PCV-2 is the main pathogen causing porcine circovirus-related diseases, which lead to significant financial losses in major swine-producing countries worldwide; these losses are associated with high swine mortality rates due to down regulation of hosts’ immune system, which, in its turn, leads to infection worsening and enables the replication of other pathogens [2,3,4,5].

PCV-2 is often diagnosed based on the assessment of clinical signs in association with antigen or nucleic acid identification through tests-based immunofluorescence, hybridization in situ, immunoperoxidase, and PCR or quantitative PCR (qPCR), which are the gold-standard methods used for porcine circovirus 2 detection purposes [6,7,8,9,10]. A TaqMan rtPCR assay was recently developed to help detecting porcine circovirus 4. Results shown that the established method was specific for PCV-4; it recorded 99.6% efficiency and detection limit of 2.2 × 10 DNA copies [11]. Liu et al. developed a sensitive assay using droplet digital PCR (ddPCR) to detect PCV-3. The detection limit recorded for ddPCR was 1 copy/mL, which was 10 times higher than that of rtPCR [12]. However, equipment cost, the need of skilled technicians to operate it, and the time required to detect PCV-2 are their main limitations. None of the vaccines and control strategies available against this infection is 100% effective so far; thus, it is essential to identify it as fast as possible in order to help prevent the disease from spreading. Therefore, next-generation technologies should present a reliable, single-step target-virus detection mechanism and require minimum detection time. The limitation of these traditional tests lies in the need of having qualified professionals and specific equipment, which are often expensive and less applicable in resource-limited environments, such as small farms [13,14].

Localized Surface Plasmon Resonance (LSPR) associated with noble metal nanostructures creates sharp spectral absorption and scattering peaks as well as strong electromagnetic near-field enhancements. The past decade has witnessed significant improvements in the production of noble metal nanostructures, which led to advancements in several biosensor and technology fields [15]. Gold nanoparticles’ (AuNPs) synthesis is among the easiest and most stable processes applied to nanomaterials. AuNPs size ranges from 1 nm to 100 nm; besides this, they have unique optical, physicochemical, and electronic properties that turn them into the ideal material to be used in LSPR biosensor production processes. They present LSPR properties that become evident when the incident light interacts with them since the light electromagnetic field leads to oscillations in the collective electron charge confined in their surface. This phenomenon can be visualized in the ultraviolet–visible (UV–Vis) region when a shift in absorbance peak at a certain wavelength changes nanoparticles’ color and enables producing colorimetric diagnostic kits [16,17]. Colorimetric biosensors for rapid testing have been increasingly traded in recent years, mainly due to features such as naked-eye determination and quick detection, which do not require complicated instruments and have low cost [18,19]. Among the different analyte types detected by these biosensors, one finds metallic cations as well as small molecules, such as DNA, antibodies, proteins, and viruses belonging to different families [20,21,22,23]. They show high potential for application in biosensing because this methodology can be induced in the infrared region by adjusting nanoparticles’ shape and size [17]. Thus, the LSPR technique for virus detection can be an interesting alternative to be adopted for PCV-2 detection. Immunomagnetic carriers can be advantageously used to isolate and concentrate target molecules and pathogens [24] in order to eliminate matrix interferences.

A new PCV-2 diagnosis methodology based on the combination of gold nanoparticles’ optical properties to magnetic properties of surface active maghemite nanoparticles (SAMNs) was herein proposed. The aim of the present study was to introduce a novel, reliable, fast, and easy-to-interpret colorimetric kit in real-world scenarios.

## 2. Materials and Methods

### 2.1. Chemicals

Gold (III) chloride trihydrate ((HAuCl_4_) 99%); sodium citrate dehydrate (99%); 3-mercaptopropiomicacid ((MPA) 99%); 11-mercaptoundecanoid acid ((MUA) 95%); N-hydroxysuccinimide (NHS); N-(3-dimethylaminopropyl)-N′-ethylcarbodiimide hydrochloride (EDC); absolute ethanol (99%); and phosphate-buffered saline (PBS, pH 7.4) were purchased at Sigma-Aldrich (Saint Louis, MO, USA).

Paramagnetic nanoparticles in the form of surface active maghemite nanoparticles (γ-Fe_2_O_3_) were provided by our collaborator Professor Fabio Vianello from University of Padova- Italy; they comprised iron (III) chloride hexahydrate (97%), nicotinamide adenine dinucleotides (NAD^+^ and NADH), 3-aminopropionaldehyde diethylacetal (APAL), ammonia solution (35% in water), and sodium borohydride (NaBH_4_), obtained from Sigma-Aldrich (Milano, Italy). The water used in the current study was obtained in a Millipore unit (Burlington, NJ, USA).

### 2.2. Antibodies

Antibodies used in the experiments were produced by our research group, which used a female rabbit immunized with 50 µg/mL (*v/v*) of purified PCV-2 [8]. The purified IgG protein concentration of 8 mg/mL was herein adopted by using bovine serum albumin (BSA) as protein standards based on the bicinchoninic acid (BCA) methods. All immunization and purification protocols were performed as described by Johnstone and Thorpe [25]. Antibodies’ production was approved by the Ethics Committee on Animal Use of FMVZ, São Paulo State University-Botucatu Campus (Protocol n. 103/2010-CEUA). 

### 2.3. Samples and Quantitative PCR (qPCR) 

Samples used in the current study derived from serum collected from swine and provided by the Molecular Diagnostic Laboratory at São Paulo State University-Botucatu Campus, Brazil. In total, 10 negative and 40 positive samples (≥ 10^2^ DNA copies/µL of serum) were confirmed based on the qPCR technique. DNA extraction for quantification analysis was performed based on using Illustra genomicPrep Mini Spin kit (GE Healthcare, Buckinghamshire, UK) by following manufacturer’s instructions. Nuclease-free water was used as control in each extraction procedure. Moreover, qPCRs were performed based on using the GoTaq^®^ qPCR Master Mix kit (Promega, Madison, WI, USA), as well as 0.2 µM of forward 5′-GAT GAT CTA CTG AGA CTG TGT GA and reverse 5′-AGA GCT TCT ACA GCT GGG ACA primers [26]. Reaction conditions and negative controls were based on Basso et al. [14] and Cruz et al. [27]. Standard curve was used to express the number of DNA copies by adopting standardized plasmid diluted from 10^5^ to 10^8^ copies of DNA/µL, in duplicate. Results were displayed in log_10_, as shown in Table 1, to help better visualizing qPCR quantification.

### 2.4. Equipment

Sample analyses were performed in Biochrom Lira S11 Ultraviolet–Visible (UV–Vis) spectrophotometer (Biochrom Ltd., Cambridge, UK). Spectra were obtained by using glass cuvette at wavelength ranging from 380 nm to 800 nm. Deionized water was used before each measurement to find the reference spectrum.

Gold and iron nanoparticles, as well as hybrid nanomaterial formation, were featured based on the transmission electron microscopy (TEM) technique, in Tecnai Spirit equipment (FEI Company, Hillsboro, OR, USA). Mean nanoparticles’ diameter was calculated in ImageJ software (version 1.8.0_172, Bethesda, MD, USA).

The qPCR equipment 7500 Fast Real Time PCR System, Thermo Fisher Scientific (Foster City, CA, USA) was used to quantify samples used in the current study.

### 2.5. Synthesis of Surface Active Maghemite Nanoparticles (SAMNs)

Patented SAMN synthesis was developed by our collaborators at University of Padova—Italy, and extensively described by Magro et al. [28,29]. Unlike other analogue iron oxide nanomaterials, SAMNs have unique properties such as high colloidal stability in the absence of any surface modifier, peculiar electro catalytic and optical properties, as well as the ability to selectively and reversibly bind proteins, peptides, and organic molecules of pharmaceutical and nutraceutical interest [30]. Nanomaterial featuring which comprised Fourier-transform infrared spectroscopy (FTIR), ^57^Fe zero-field and in-field Mössbauer spectroscopy, magnetization measurements, and x-ray diffraction was reported elsewhere [31,32]. Appendix A presents the TEM micrograph of SAMNs. Besides the homogenous spherical morphology, samples were featured by significant monodispersity; mean SAMN diameter was calculated and resulted in 13.72 ± 1 nm (Appendix A).

### 2.6. AuNPs’ Preparation

AuNPs were prepared based on the method described by Basso et al. [33]. The aliquot of 10 mL HAuCl_4_ (1 mM) was added to Erlenmeyer glass until it reached 100 °C. Then, 2 mL of sodium citrate (10 mM) was added to the solution, whose color changed into red right away; this outcome has indicated Au^3+^ complex reduction to Au^0^, which led to nanoparticles’ formation. The final AuNPs solution has showed concentration of 0.1 g L^−1^. TEM images have shown homogeneous and spherical nanoparticles (Appendix A), whose mean diameter reached 23.89 ± 0.25 nm (Appendix A). 

### 2.7. SAMNs/AuNPs Hybrid Preparation (SAMN@MPA@AuNPs)

Mercapto-carboxylic acids are elective coupling agents used to crosslink SAMNs to gold surfaces. The chelating properties of carboxyl groups enable anchoring the organic moiety to the iron oxide surface based on a simple self-assembly wet reaction. Consequently, –SH groups are exposed to the solvent and available to bind a gold surface, as substantiated by FTIR and quartz crystal microbalance in our previous study [32]. Thus, a binary hybrid was herein prepared based on the combination between SAMNs and AuNPs, by using 3-mercaptopropiomic acid (MPA) as cross linker in order to develop an immunomagnetic colorimetric biosensor. First, 0.1 g L^−1^ of SAMN aqueous suspension was subjected to magnetic separation (Appendix A); then, water was replaced by 1 mM MPA solution and incubated at room temperature for 40 min. After the incubation period was over, MPA-modified SAMNs (SAMN@MPA) were magnetically retrieved to remove MPA excess. Then, deionized water was used to wash the nanoparticles and to remove the unbound MPA; this process was repeated three times. The as-obtained SAMN@MPA was added to 0.1 g L^−1^ of AuNPs water suspension and incubated at room temperature for 40 min to form the SAMN@MPA@AuNp hybrid.

## 3. Results and Discussion

### 3.1. Determining the Ideal γ-Fe_2_O_3_ Nanoparticles’ Concentration

First, γ-Fe_2_O_3_ concentrations of 0.1, 0.2, 0.3, 0.4, and 0.5 g L^−1^ were tested to determine the ideal γ-Fe_2_O_3_ nanoparticles’ concentration to be bound to AuNPs and form the hybrid nanomaterial. These values were based on previous publications [28,30]. Concentrations of 0.5, 0.4, 0.3, and 0.2 g L^−1^ have shown noisy signals and undefined peaks. Spectra recorded at concentration of 0.1 g L^−1^ have shown noiseless and more defined peaks, with absorbance of 0.47, at wavelength of 511.28 nm. This concentration (0.1 g L^−1^) enabled high colloidal stability means low to aggregate, which led to high surface available for the interaction with MPA and gold nanoparticles. This is the reason why it was selected to be used in the following experiments (Figure 1).

### 3.2. Preparing the Immunomagnetic Colorimetric System

It was necessary modifying the surface of hybrid nanoparticles so they could bind specific PCV-2 antibodies in order to produce the immunomagnetic colorimetric biosensor. The adopted modification steps were featured based on Ultraviolet–Visible (UV–Vis) graphs in spectrophotometer (Figure 2A).

The adopted γ-Fe_2_O_3_ presented absorbance peak of 0.47 and wavelength of 511.28 nm (black line) before hybrid formation. Bare AuNPs have shown absorbance peak at 524.92 nm (red line); this result was compatible to those described in the literature [33]. Nanoparticles’ binding based on the use of MPA solution (described in item 2.7) produced a γ-Fe_2_O_3_/AuNPs hybrid nanoparticle. These nanoparticles present absorbance peak of 0.94 at 540.58 nm (green line). Previously study conducted by our research group has shown that using this hybrid complex has increased colorimetric detection sensitivity to dengue virus due to nanoparticles’ dielectric properties [32]. The surface of the formed hybrid was modified with 11MUA solutions to enable it to bind to PCV-2 antibodies because it has a thiol group capable of binding to AuNPs’ surface; it leaves the carboxylic group at the other end of it free. The 11MUA (0.018 mol L^−1^) addition has changed the wavelength shift of the hybrid complex to 544.32 nm (blue line). The free carboxylic group was activated at 1:1 solution comprising EDC (0.1 mol L^−1^) and NHS (0.05 mol L^−1^) in order to create covalent bonds to antibodies’ amines. There was change in absorbance peak to 0.90 due to shift in wavelength to 554.36 nm (pink line) at this step. The naked γ-Fe_2_O_3_ showed a yellow-earth color at this stage of the work, whereas the naked AuNPs presented red color, the γ-Fe_2_O_3_/AuNPs hybrid showed orange color, and after modification with MUA and EDC-NHS, purple color (Figure 2B).

### 3.3. Colorimetric Analysis of Dilutions

PCV-2 antibodies were added to the solution after modifications were carried out on the surface of hybrid nanoparticles to enable covalent binding between antibody amines and nanoparticles. The herein-used antibodies concentration was 2.5 µg/mL (30 µL) based on previous study by our group; the color of the solution remained purple [14]. The first part of the analyses was carried out to find the best dilution to be adopted. Negative and positive samples with log 2 at 5 were tested at the following concentrations: pure (no dilution), 1:10, 1:100, 1:1000, and 1:10,000 (30 µL in PBS buffer). 

Dilution steps were analyzed in spectrophotometer; UV–Vis graphs have evidenced absorbance peak and wavelength shift at each tested concentration. The SO3G3 sample was used in log_2_ dilutions. Absorbance peaks at this concentration (log_2_) ranged from 0.94 to 1.28 at wavelength ranging from 578.70 to 601.89 nm. Colorimetric change in the developed biosensor was also observed at this dilution step. Pure sample (no dilution) addition showed red color similar to that observed after negative sample addition; thus, there was no significant difference between them to the point that they could be easily mistaken for one another. It happened due to the prozone effect since excess of virus in the tested serum has affected the antigen–antibody complex formation. The other dilutions maintained the purple hue, which enabled differentiating them from the negative sample to the naked eye (Figure 3A). The SO10774 sample was used in log_3_ dilutions, at absorbance peaks ranging from 0.94 to 1.23, and at wavelength ranging from 593.07 to 646.33 nm. Colorimetric analysis also evidenced that pure and negative samples showed red color, whereas the other dilutions showed purple color (Figure 3B). The SO5G1 sample was used for log_4_. Maximum absorbance peak reached 1.23 at wavelength ranging from 617.22 to 644.43 nm; it presented purple color in the dilutions and red color in negative samples (Figure 3C). SO8802 with log_5_ was the last tested positive sample. This sample showed peaks ranging from 0.84 to 1.14 in different dilutions, at wavelength ranging from 634.30 to 646.46 nm. The color pattern remained the same as that in the previous samples, namely red for negative samples and purple for positive samples. (Figure 3D). On the other hand, negative sample dilutions did not show shift in peak wavelength in comparison to the antibody. Absorbance values ranged from 544.93 to 592.30 nm, and the antibody peaked at 597.06 nm; this outcome showed lack of binding between the virus and hybrid nanoparticles modified with antibodies. Unlike the positive samples, all negative sample dilutions were red, a fact that evidenced lack of binding between the complex and the virus (Figure 3E).

Based on the comparison of all dilutions performed in the analyzed samples, visual limit of detection (LOD) and UV–Vis spectrum occurred in 1:1000 dilutions. The log_2_ sample at the 1:1000 dilution showed a absorbance peak at 601.47 nm and the 1:10,000 dilution at 601.89 nm. The log_3_ sample was at 646.10 and 646.33 nm; log_4_ sample at 644.01 and 644.43 nm; log_5_ sample at 647.44 and 646.46 nm for the 1:1000 and 1:10,000 dilutions, respectively. The analysis applied to the herein-recorded values enabled seeing that all samples at 1:100 dilution presented shift in wavelength in comparison to the antibody. This outcome indicated binding between the PCV-2 virus and the whole complex; it was similar to results reported in the literature by Basso et al. [34]. It was possible to observe that the 1:100 dilution enabled the best color differentiation from that of the negative sample as well as that it resulted in the best and most efficient antigen–antibody binding. Absorbance peaks observed at 1:100 dilutions were also more defined and less noisy, which was the reason why this dilution was selected as the ideal one to be used in the other experiments.

### 3.4. PCV-2 Detection

In total, 10 negative and 40 positive samples (ranging from log_2_ to log_5_) were tested under optimal conditions, as shown in Table 1. One (1) positive sample and one negative sample from each log are shown in the UV–Vis graphs to help better visualizing the results. Results concerning the other samples are shown in Appendix A. All modifications and bonds on nanoparticles’ surface were presented based on the LSPR technique, with emphasis on their wavelength shift. Figure 4A shows the detection of positive sample SO6G3-log_2_. The black line represents the formation of hybrid nanoparticles modified with self-assembled monolayers in extinction peak at 558.28 nm. There was shift in wavelength to 593.07 nm (red line) after antibodies’ addition. There was shift in the absorbance peak to 599.31 nm after sample addition, a fact that indicated antigen–antibody binding (blue line). The other nine samples with log_2_ showed changes ranging from 594.10 to 605.12 nm (see Appendix A).

Sample SO10779 was displayed in Figure 4B for log_3_. Nanoparticles showed absorbance peak at 540.88 nm (black line). Nanoparticles’ surface modification with antibodies showed an absorbance peak at 581.18 nm (red line), and after sample’s addition, the peak shifted to 640.34 nm (yellow line). This displacement took place due to increase in local refractive index on the surface of the complex, which was caused by antigen–antibody binding. The other results of the connections ranged from 640.21 to 643.99 nm (see Appendix A). The absorption value of 1.04 at 584.0 nm in the red line of the Figure 4C indicated anti-PCV-2 antibodies. Absorption value changed to 0.77 at 646.54 nm (pink line) after the SO11974-log_4_ sample injection. The other samples of the same log recorded absorbance peaks ranging from 641.84 to 646.12 nm when they bound to antibodies (see Appendix A). With respect to the sample with log_5_ (Figure 4D), only hybrid nanoparticles showing self-assembled monolayers’ formation on their surface recorded an absorbance peak of 0.99 at 542.77 nm (black line). The covalent binding of antibodies on hybrid surface lead to drop in absorbance peak to 0.94 as well as to shift in wavelength to 568.49 nm (red line). Antigen–antibody binding took place after SO8795 sample addition, and the absorbance peak shifted to 644.38 nm at absorbance of 0.92 (orange line). Absorbance peaks observed after the other samples bound to antibodies ranged from 645.12 to 648.12 nm (see Appendix A).

On the other hand, there was no shift in wavelength when negative samples were added at hybrid complex. Antibodies’ addition showed absorbance peak at 596.36 nm; after negative sample addition, absorbance peak was observed at 589.05 nm (Figure 4E, green line). Peaks observed for other samples ranged from 587.41 to 592.30 nm; this outcome evidenced lack of antigen–antibody binding (see Appendix A). The most significant wavelength shifts between samples were observed from log_3_ to log_5_. Samples with log_2_ bound to antibodies although the observed wavelength shifts were not large, a fact that was justified by lower virus concentration in animals’ sera. Wavelength shifts take place due to virus and antibodies’ agglutination with hybrid nanoparticles, a fact that reduces their surface area and, consequently, decreases plasmon resonance [35]. Binding of positive samples from all logs to antibodies can be observed by the naked eye since positive samples turn purple, and negative samples turn red. The kit color-change process took approximately 1 min after sample addition in hybrid complex; it is a fast and easy process that can also be measured in a quantitative manner with the aid of ultraviolet–visible spectrophotometer. Figure 5 shows color changes observed for positive (SO6G3-log_2_ and SO8795-log_5_) and negative samples when they were added to the nanoparticle–antibody complex. Samples’ pH values were monitored throughout the experiment and ranged from 5.0 to 5.7. The complete Appendix A.

### 3.5. Transmission Electron Microscopy and Negative Controls

In addition to UV–Vis graphs and colorimetric visualization, nanoparticles’ morphological featuring was performed based on the transmission electron microscopy technique and analyzed in ImageJ software. As previously mentioned, γ-Fe_2_O_3_ without modifications showed mean size 13.72 nm, whereas the mean size observed for AuNPs was 23.89 nm (Figure 6A,B). Self-assembled monolayers were created on nanoparticles’ surface and formed a visually detectable small shadow around them after the hybrid was formed and its surface was modified by using 11MUA/EDC-NHS. PCV-2 antibodies were then added at hybrid complex, and nanoparticles reached mean diameter 45.2 nm. These results are consistent with studies conducted by Reth [36] (Figure 6C). There was visible change in nanoparticles’ shape after positive serum sample (SO8802 log_5_) addition. Darkly stained centers with defined light edges and icosahedral symmetry were observed in nanoparticles. These images are similar to those found by Gava et al., who described the structural insight in the type-specific epitope of porcine circovirus [37]. Nanoparticles started to present mean diameter 76.61 nm, which was consistent with the size of the PCV-2 virus, i.e., 17 nm [38] (Figure 6D).

In addition to 10 negative serum samples collected from different animals likely interferences in similar structural proteins, which can cause false-positive diagnoses in conventional tests, were also tested to validate the experiment. Porcine circovirus has four species, namely PCV-1, PCV-2, PCV-3, and PCV-4. PCV-1 is nonpathogenic, but it is often found in animals and can be mistaken by PCV-2; thus, it was used to evaluate experimental specificity and selectivity towards antibodies. In addition, non-enveloped DNA viruses, similar to PCV-2, belonging to families *Adenoviridae* and *Parvoviridae*, were also analyzed [37,39,40]. Figure 7 shows hybrid nanoparticles modified through self-assembling monolayers and antibodies in the black line; absorbance peak of 1.04 was observed at 598.94 nm. There was no shift in absorbance peak after the addition of PCV-1-positive serum sample; the red line on the graph shows the value of absorbance peak at 593.59 nm. The same pattern was observed for adenovirus, which recorded absorbance peak of 1.06 at 592.8 nm (green line) as well as for parvovirus, which recorded absorbance peak of 0.84 at 594.59 nm. The lack of absorbance peak displacement when interferents were added indicated lack of binding between the virus and antibodies, a fact that confirmed its specificity to PCV-2. Moreover, no colorimetric changes were observed after the addition of interfering samples, a fact that ruled out likely false-positive results.

## 4. Conclusions

The current study developed a colorimetric kit for rapid PCV-2 virus detection based on using hybrid γ-Fe_2_O_3_- AuNPs nanoparticles. The surface of hybrid nanoparticles was modified by the formation of self-assembled monolayers for antibody coupling and complex-formation purposes. Serum samples collected from pigs with and without PCV-2 virus were tested to validate and produce the aforementioned kit. Color change took place within approximately 1 min after samples’ addition at hybrid complex; positive samples turned purple, whereas negative samples turned red. These changes could be seen by the naked eye without using any specific equipment. The final cost of the colorimetric kit was USD 4.00 per sample. Its low cost and easily interpreted results turned it into an interesting alternative to traditional detection tests. Moreover, the present study can be used as basis to produce new biosensors and to detect other virus and bacterium types.

## Figures and Tables

**Figure 1 pathogens-11-00570-f001:**
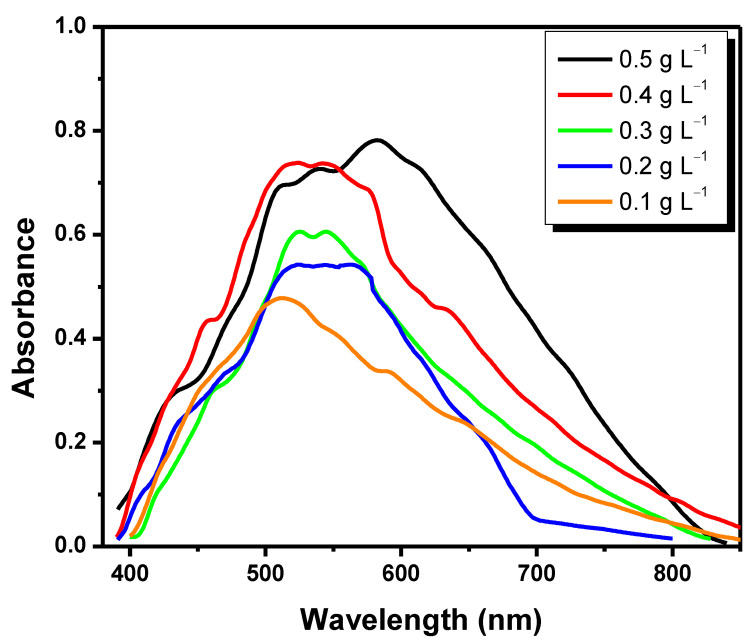
Ultraviolet–Visible (UV–Vis) graph for different γ-Fe_2_O_3_ concentrations.

**Figure 2 pathogens-11-00570-f002:**
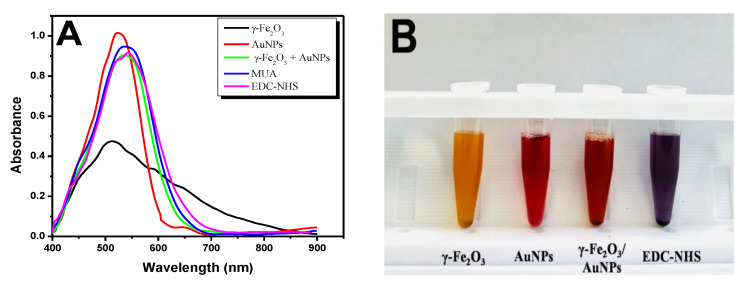
Hybrid nanoparticles and surface modifications. (**A**) Spectrum showing γ-Fe_2_O_3_ nude (black line), AuNPs nude (red line), hybrid nanoparticles (green line), 11-mercaptoundecanoid acid ((11MUA) blue line), and N-hydroxysuccinimide-N-(3-dimethylaminopropyl)-N′-ethylcarbodiimide hydrochloride ((EDC-NHS) pink line). (**B**) Colorimetric changes recorded for the first modifications. Γ-Fe_2_O_3_ nude (yellow color), AuNPs nude (red color), γ-Fe_2_O_3_/AuNPs hybrid (orange color), and modification with MUA/EDC-NHS (purple color).

**Figure 3 pathogens-11-00570-f003:**
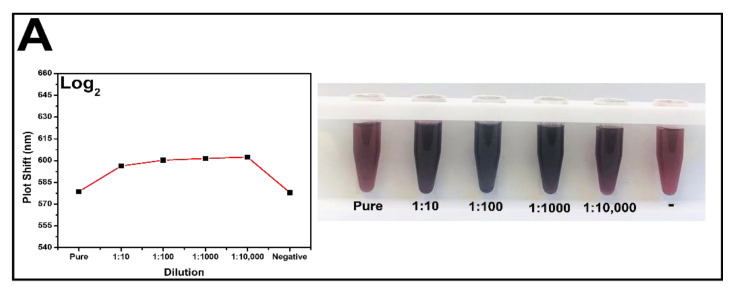
Dilution of different sample concentrations and colorimetric change. (**A**) SO3G3 log_2_ sample; (**B**) SO10774 log_3_ sample; (**C**) SO5G1 log_4_ sample; (**D**) SO8802 log_5_ sample; and (**E**) negative sample.

**Figure 4 pathogens-11-00570-f004:**
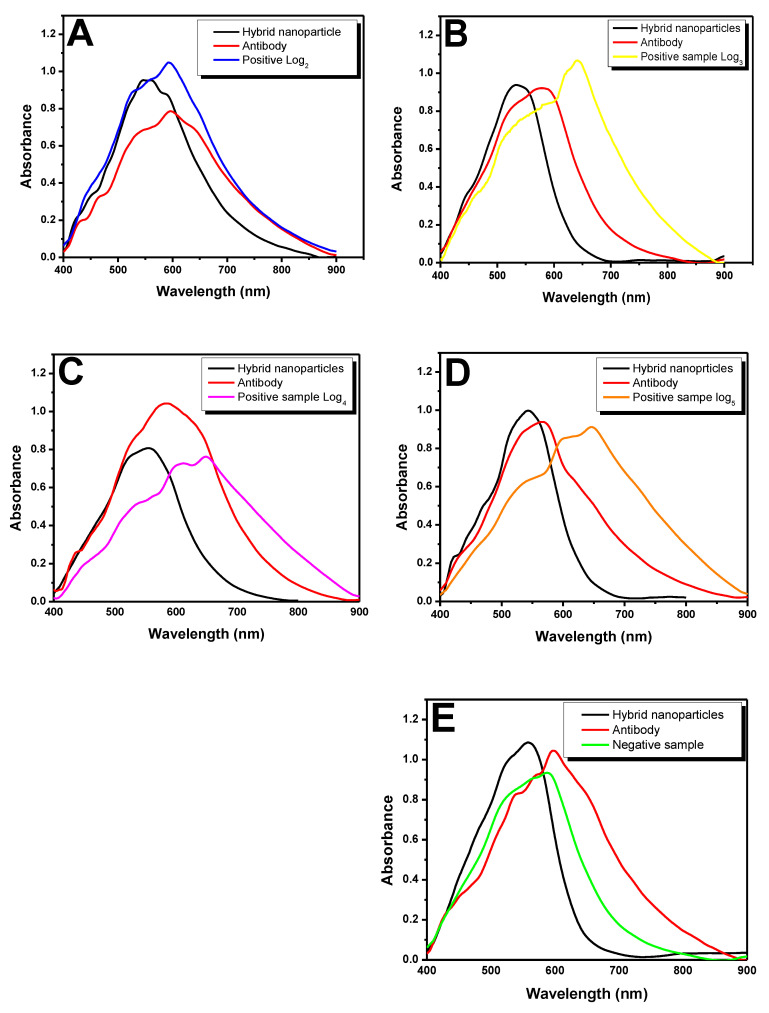
UV–Vis spectrum showing positive samples binding to antibodies immobilized on the surface of hybrid nanoparticles. (**A**) SO6G3 log_2_ sample; (**B**) SO10779 log_3_ sample; (**C**) SO11974 log_4_ sample; (**D**) SO8795 log_5_ sample; and (**E**) negative sample.

**Figure 5 pathogens-11-00570-f005:**
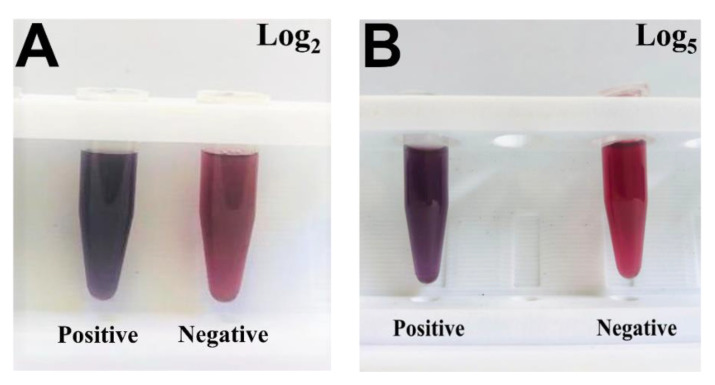
Colorimetric change. (**A**) SO6G3-log_2_ sample (purple) and negative (red) sample. (**B**) SO8795 log_5_ sample (purple) and negative (red) sample application.

**Figure 6 pathogens-11-00570-f006:**
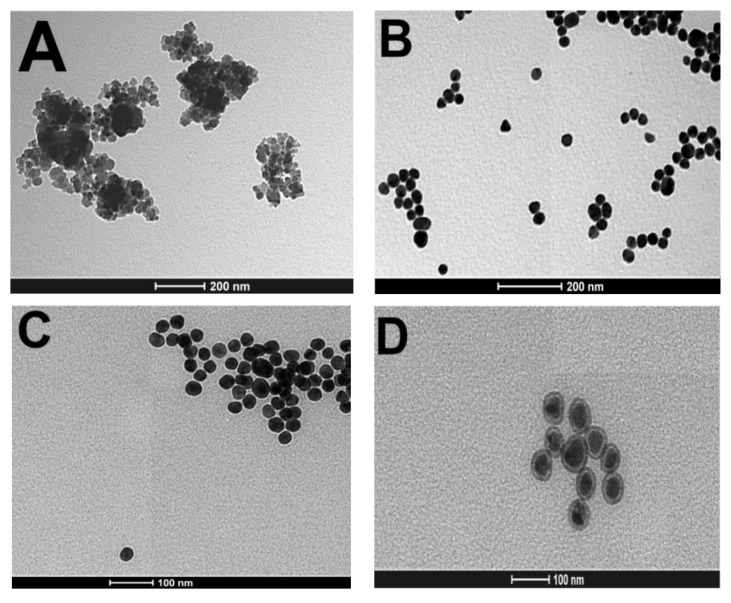
TEM images showing nanoparticles’ surface modification steps. (**A**) γ-Fe_2_O_3_ nude with mean size 13.72 nm, at 200 nm scale. (**B**) AuNPs nude with mean size 23.89 nm, at 200 nm scale. (**C**) Hybrid nanoparticles and surface modification using 11MUA/EDC-NHS with mean size 45.20 nm, at 100 nm scale; and (**D**) positive sample binding to the antibody with mean size 76.61 nm, at 100 nm scale.

**Figure 7 pathogens-11-00570-f007:**
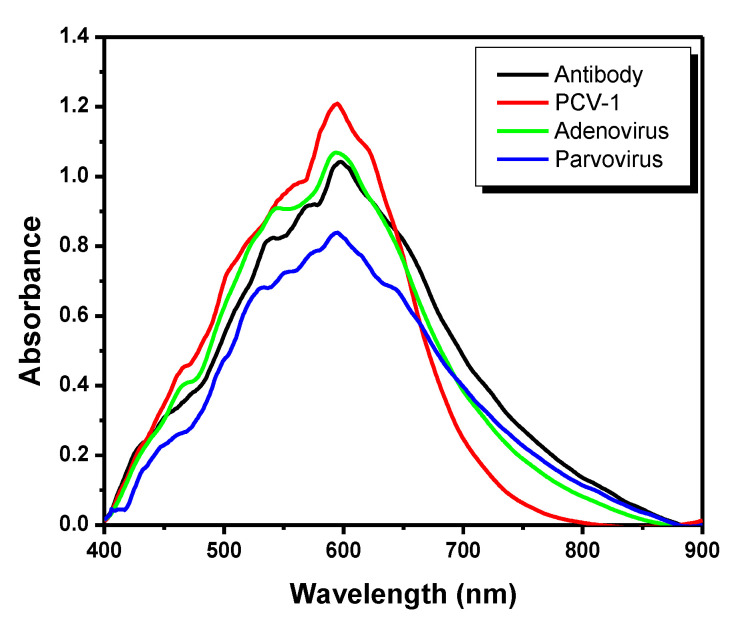
UV–Vis graph used in the analysis of interfering samples. Antibody (black line), PCV-1 (red line), adenovirus (green line), and parvovirus (blue line).

**Table 1 pathogens-11-00570-t001:** Sample quantification based on quantitative PCR (qPCR), expressed as the number of porcine circovirus 2 (PCV-2) DNA copies/µL.

Sample Identification	Quantification (Copies/µL)	Log_10_
SO12178 (1)	102.76 (1.0 × 10^2^)	2.01
SO2G3 (2)	243.53 (2.4 × 10^2^)	2.39
SO10754 (3)	261.17 (2.6 × 10^2^)	2.41
SO3G3 (4)	258.51 (2.5 × 10^2^)	2.41
SO12669 (5)	372.85 (3.7 × 10^2^)	2.57
SO10765 (6)	421.87 (4.2 × 10^2^)	2.62
SO12166 (7)	427.69 (4.2 × 10^2^)	2.63
SO11948 (8)	524.12 (5.2 × 10^2^)	2.71
SO12670 (9)	744.71 (7.4 × 10^2^)	2.87
SO6G3 (10)	924.35 (9.2 × 10^2^)	2.97
SO8783 (11)	1053.00 (1.0 × 10^3^)	3.02
SO10768 (12)	1048.94 (1.0 × 10^3^)	3.02
SO12659 (13)	1433.95 (1.4 × 10^3^)	3.15
SO10782 (14)	1606.38 (1.6 × 10^3^)	3.20
SO8791 (15)	2654.20 (2.6 × 10^3^)	3.42
SO10767 (16)	3638.10 (3.6 × 10^3^)	3.56
SO8808 (17)	4194.48 (4.1 × 10^3^)	3.62
SO8786 (18)	5222.79 (5.2 × 10^3^)	3.71
SO10774 (19)	5320.76 (5.3 × 10^3^)	3.72
SO10779 (20)	6998.65 (6.9 × 10^3^)	3.84
SO8819 (21)	10,021.12 (1.0 × 10^4^)	4.00
SO8777 (22)	15,282.53 (1.5 × 10^4^)	4.18
SO7G1 (23)	15,617.45 (1.5 × 10^4^)	4.19
SO8797 (24)	25,536.00 (2.5 × 10^4^)	4.41
SO8803 (25)	31,019.83 (3.1 × 10^4^)	4.49
SO8798 (26)	48,457.79 (4.8 × 10^4^)	4.68
SO8823 (27)	54,500.54 (5.4 × 10^4^)	4.73
SO5G1 (28)	63,113.80 (6.3 × 10^4^)	4.80
SO8810 (29)	67,447.09 (6.7 × 10^4^)	4.82
SO11974 (30)	88,895.16 (8.8 × 10^4^)	4.94
SO12479 (31)	11,2631.20 (1.1 × 10^5^)	5.05
SO8820 (32)	112,741.70 (1.1 × 10^5^)	5.05
SO12468 (33)	124,190.49 (1.2 × 10^5^)	5.09
SO8799 (34)	128,539.20 (1.2 × 10^5^)	5.10
SO8800 (35)	151,231.50 (1.5 × 10^5^)	5.17
SO8790 (36)	186,290.20 (1.8 × 10^5^)	5.27
SO8788 (37)	241,207.80 (2.4 × 10^5^)	5.38
SO8802 (38)	377,181.30 (3.7 × 10^5^)	5.57
SO8787 (39)	824,489.50 (8.2 × 10^5^)	5.91
SO8795 (40)	909,932.00 (9.0 × 10^5^)	5.95

## Data Availability

Not applicable.

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
