# Peer review of "Colorimetric Kit for Rapid Porcine Circovirus 2 (PCV-2) Diagnosis"

_pathogens, 2022, doi:10.3390/pathogens11050570_

Round 1
Reviewer 1 Report
Manuscript entitled „ Colorimetric Kit for Rapid Porcine Circovirus 2 (PCV-2) Diagnosis” is very interesting, well-written and well-planned experimental work. Small corrections should be made to the text according to the following comments:
Introduction
line 42 – explain in full name abbreviation PCV
line 42,43 – use abbreviation rtPCR after TagManPCR
line 47 – use only abbreviation rtPCR without the full name explanation
line 62 – use only abbreviation AuNPs
line 64 – explain in full name for the first time abbreviation LSPR
line 65 – use only abbreviation SPR instead of full name
Materials and Methods
line 88 – put space between coma and 99%
line 88 - write in the same order as before first abbreviation (MUA) and later concentration 95%
line 97 – write Sigma-Aldrich (Italy)
line 102 – explain in full name abbreviation BSA
Table1 – explain in full name abbreviations qPCR and PCV-2 which for the firs time are used in the description of the table; some number positions use commas, replace them with dots
line 144 – explain in full name abbreviation FTIR
line 146 – live only abbreviation TEM instead of full name
Figure 1 and Figure 2 - explain all the abbreviations that first appeared in the figures
Results and Discussion
Figure 4 – explain abbreviation UV-Vis for the first time
line 204 – explain in full name UV-Vis
Figure 5 - all abbreviations that were not previously used in the description of tables or figures must be explained with the full name
line 239 - replace comma with a dot
line 290 – use only abbreviation LSPR its full name was explained previously in the text
References
bind all items of literature in accordance with the template applicable in the journal
Journal Articles:
- Author 1, A.B.; Author 2, C.D. Title of the article. Abbreviated Journal Name Year, Volume, page range.
Reviewer 2 Report
In the article entitled: "Colorimetric Kit for Rapid Porcine Circovirus 2 (PCV-2) Diagnosis" authors provided an interesting and well-designed study regarding the rapid and costless detection of ubiquitous PCV2-virus.
The study provided by authors is interesting, introduction is sufficient, methods are sound, results are clearly described.
Major comments:
- Please provide photography of log2-sample in comparison to negative control in the figure 8, from the figure 6 we may know that the colour will be probably the same, but this would be helpful.
- Please provide the information about limit of detection (LOD). In the Figure 6 we may see there is still color-change in 1:10 000 dilution of log2 sample (even better plot shift). It means that sample containing 100 DNA copies/µl is still detectable when diluted to 1 DNA copy/0,01 µ That is quite low and unusual, much lower than qPCR can provide. Please comment that.
Minor comments:
- Line 57: LSPR
- Line 288: S1-S3, there is no S4 and S5 available in supplementary file
- Please adjust the figures, many of them are in different sizes, lower or higher. Please use the same size. Some of them may be moved to supplementary files (i.e Fig 1-3).
I have no further comments.
